# Spaghetti to a Tree: A Robust Phylogeny for Terebelliformia (Annelida) Based on Transcriptomes, Molecular and Morphological Data

**DOI:** 10.3390/biology9040073

**Published:** 2020-04-06

**Authors:** Josefin Stiller, Ekin Tilic, Vincent Rousset, Fredrik Pleijel, Greg W. Rouse

**Affiliations:** 1Scripps Institution of Oceanography, University of California, San Diego, CA 92037, USA; etilic@evolution.uni-bonn.de (E.T.);; 2Centre for Biodiversity Genomics, Department of Biology, University of Copenhagen, 2100 Copenhagen, Denmark; 3Institute of Evolutionary Biology and Animal Ecology, University of Bonn, 53121 Bonn, Germany; 4Tjärnö Marine Laboratory, Department of Marine Sciences, University of Gothenburg, 405 30 Gothenburg, Sweden; fredrik.pleijel@marine.gu.se

**Keywords:** phylogenomics, transcriptomics, Annelida, Terebelliformia

## Abstract

Terebelliformia—“spaghetti worms” and their allies—are speciose and ubiquitous marine annelids but our understanding of how their morphological and ecological diversity evolved is hampered by an uncertain delineation of lineages and their phylogenetic relationships. Here, we analyzed transcriptomes of 20 terebelliforms and an outgroup to build a robust phylogeny of the main lineages grounded on 12,674 orthologous genes. We then supplemented this backbone phylogeny with a denser sampling of 121 species using five genes and 90 morphological characters to elucidate fine-scale relationships. The monophyly of six major taxa was supported: Pectinariidae, Ampharetinae, Alvinellidae, Trichobranchidae, Terebellidae and Melinninae. The latter, traditionally a subfamily of Ampharetidae, was unexpectedly the sister to Terebellidae, and hence becomes Melinnidae, and Ampharetinae becomes Ampharetidae. We found no support for the recently proposed separation of Telothelepodidae, Polycirridae and Thelepodidae from Terebellidae. Telothelepodidae was nested within Thelepodinae and is accordingly made its junior synonym. Terebellidae contained the subfamily-ranked taxa Terebellinae and Thelepodinae. The placement of the simplified Polycirridae within Terebellinae differed from previous hypotheses, warranting the division of Terebellinae into Lanicini, Procleini, Terebellini and Polycirrini. Ampharetidae (excluding Melinnidae) were well-supported as the sister group to Alvinellidae and we recognize three clades: Ampharetinae, Amaginae and Amphicteinae. Our analysis found several paraphyletic genera and undescribed species. Morphological transformations on the phylogeny supported the hypothesis of an ancestor that possessed both branchiae and chaetae, which is at odds with proposals of a “naked” ancestor. Our study demonstrates how a robust backbone phylogeny can be combined with dense taxon coverage and morphological traits to give insights into the evolutionary history and transformation of traits.

## 1. Introduction

Terebelliformia is a clade of Annelida with considerable ecological and morphological diversity. Members can be recognized by the long and convoluted buccal palps that originate from inside or outside the mouth, which inspired their common name “spaghetti worms”. With ~1100 named species, Terebelliformia is one of the most speciose groups of annelids [1] and its species range from shallow coastal areas to the deep sea, occupying habitats ranging from mud flats, coralline reefs, whale falls to hydrothermal vents [2,3]. Their diversity is likely underestimated as new species are continually being discovered [4,5]. An illustration of the underappreciated biodiversity is the “cosmopolitan” species *Terebellides stroemii* that was recently shown to comprise 27 species in the northeast Atlantic alone [6]. Most terebelliforms are tube-dwelling and their bodies have been transformed to build, and live inside, a tube. The tube mucus is secreted from ventral glandular areas and the worm uses the lips on its head to adorn the tube with shells, clay and sand grains [7]. While keeping its body safely anchored in the tube with specialized neuropodial chaetae, the animal only exposes its branchiae to absorb oxygen and its palps to feed. Terebelliforms show remarkable variation in palps, branchiae and chaetae (Figure 1). The combination of these features has traditionally been used to delineate the major lineages [8,9]: Terebellidae Grube, 1850 (including Terebellinae Hessle, 1917, Polycirrinae Malmgren, 1866, Thelepodinae Malmgren, 1866), Trichobranchidae Malmgren, 1866, Pectinariidae Quatrefages, 1866, Ampharetidae Malmgren, 1866 (including Ampharetinae Malmgren, 1866, Melinninae Chamberlin, 1919) and Alvinellidae Desbruyères and Laubier, 1986. Understanding how this diversity arose requires a phylogenetic hypothesis with robust support and, ideally, with broad taxonomic sampling.

Phylogenomics have confidently placed Terebelliformia within Annelida as the sister group to Maldanomorpha (Arenicolidae+Maldanidae) [8,9,10], but sampling has not been comprehensive enough to inform relationships within Terebelliformia. Six species were sequenced for mitochondrial genomes and three nuclear genes but this only represented a fraction of the group’s diversity [11]. Better taxonomic coverage was achieved using single or few molecular loci or morphological data, but these usually failed to provide confidently resolved hypotheses [12,13,14,15,16]. In fact, nearly every possible combination of relationships among the main lineages has been proposed. The latest cladistic analysis of 118 morphological traits inferred that much of the traditional delimitation of Terebelliformia had been mistaken because Terebellidae included all other Terebelliformia [13]. Despite not evaluating statistical support for the proposed terebellid paraphyly, a new classification was established with new ranks for lineages that had been traditionally part of Terebellidae, namely Thelepodidae, Polycirridae and Terebellidae sensu stricto, as well as a new family, Telothelepodidae Nogueira, Fitzhugh and Hutchings, 2013 [13]. 

The relationships among the other terebelliform groups also remain unclear. Even the placement of the iconic Pompeii worms (Alvinellidae) is not resolved. Upon their discovery on hydrothermal vents, Pompeii worms were considered to be Ampharetidae because they share the ability to fully retract their buccal palps, before being given family rank [17,18]. Early molecular studies failed to confidently place them [16,19,20]. Other data supported a sister group relationship between Alvinellidae and Ampharetinae (though no Melinninae included) [11,14], or inclusion within Ampharetidae with Melinninae being the sister to Ampharetinae+Alvinellidae [2,21]. Pectinariidae have been proposed as the sister group to all other Terebelliformia [11,22], but have also been placed within Ampharetidae [13], or even outside of Terebelliformia [16]. Trichobranchidae have been treated as members of Terebellidae based on morphological similarities [23] or as a separate lineage [14,24], while molecular analyses have placed them as the sister to Alvinellidae [22] or found them to be paraphyletic [16].

In addition to the significance to systematics, different phylogenetic hypotheses of terebelliform relationships alter the interpretation of their trait evolution, for example of the branchiae. Branchiae show an unusual variety and can occur in one to six pairs, can be fused into a single stalk, split into filaments, or can be shaped like a comb, a tree or a bottlebrush [25,26] (Figure 1). Polycirridae lack branchiae altogether and also show varying degrees of chaetal reduction, giving them a “naked” appearance. Most polycirrids are free-living [27]. They were traditionally regarded as simplified Terebellidae because they share palps inserting outside of the mouth and similar glandular shields [24,28]. The latest systematic revision found them as the sister to all other Terebelliformia, which was interpreted as evidence that the absence of branchiae may have been plesiomorphic for Terebelliformia [13]. By extension, this suggests that terebelliforms may have been ancestrally “naked” with poorly developed chaetae. The placement of Polycirridae is therefore of significance to infer polarity of morphological changes.

Here, we chose a threefold strategy to reconstruct the phylogeny of Terebelliformia and to study the evolutionary transitions of characters. First, we constructed a robust phylogenetic backbone for the main lineages grounded on a large number of genetic loci (12,674 genes, 4.6 million amino acids) from analyzing transcriptomes of 21 species of the major lineages, including an outgroup. Second, we assessed more fine-scale relationships within the main clades by constructing a phylogeny for 122 species sequenced for five Sanger-sequenced genes (three nuclear, two mitochondrial loci; 4159 bp) and scored for 90 morphological characters, which we constrained with the transcriptome-based backbone phylogeny. We then undertook a morphological review in order to identify homologous structures across all Terebelliformia by tracing the morphological characters on the resulting phylogeny. We used this approach, with genome-wide sampling on one hand and broad taxonomic resolution on the other, to better understand the origins of key traits, particularly the diversity of branchiae.

## 2. Materials and Methods

### 2.1. Transcriptome Sequencing and Bioinformatic Processing

New transcriptomes were generated for 11 species, which were selected to cover taxonomic diversity and complement 10 published transcriptomes (Appendix A). The outgroup *Arenicola marina* was chosen based on the consistent support of Maldanomorpha as sister to Terebelliformia [8,10,11,22]. We included at least one representative of each of the main lineages recognized by Holthe 1986 [25]. Of the eight terebelliform families, *sensu* Nogueira et al. 2013 [13], we include one Pectinariidae, five Ampharetidae (four Ampharetinae, one Melinninae), six Alvinellidae, two Trichobranchidae, one Polycirridae, one Thelepodidae and four Terebellidae. The only missing lineage is Telothelepodidae, for which no suitable material was available for transcriptome sequencing (but a member of the taxon was Sanger sequenced).

Tissue was either finely chopped, placed in RNAlater (Ambion) for 24 h at 4 °C and then stored at −80 °C or −20 °C, or it was flash-frozen in liquid nitrogen and stored at −80 °C. Total RNA was extracted with Trizol (Ambion) using Direct-zol RNA kits (Zymo Research) including a DNase I digest to remove genomic DNA following the manufacturer’s specifications. mRNA was purified with a Dynabeads mRNA Direct Micro Kit (Invitrogen). Quantity and quality of mRNA were assessed with a Qubit fluorometer (Invitrogen) and Tapestation RNA Screen Tape (Agilent). Libraries were prepared using KAPA Stranded mRNA-Seq kits (KAPA Biosystems) following the recommended protocol and using custom 10 nucleotide Illumina TruSeq style adapters [29]. Libraries were sequenced on the Illumina HiSeq4000 using 100 base pairs paired-end reads at the IGM Genomics Center (University of California San Diego). Samples had on average 41.9 million reads (range 28.4–54.5; Appendix A). Raw sequence data have been deposited in the NCBI sequence read archive (SRA) under Bioproject PRJNA611902 (Appendix A).

Raw reads from published transcriptomes (average: 43.7, range 4.6–163.3 million reads) were processed in the same manner as the new sequence data. Three published transcriptomes (*Alvinella caudata, Paralvinella pandorae irlandei* (now *Nautalvinella irlandei*, see below) and *Lanice conchilega*) had only 4.6–7.2 million reads but the samples were kept because their phylogenetic position was resolved with high confidence. Sequence reads were trimmed and cleaned of adapters using Trimmomatic v. 0.36 [30]. The pipeline Agalma v. 1.0.1 [31] was used for processing steps, from a second round of quality threshold filtering, de novo assembly using Trinity v. 2014-04-13 [32], translation to amino acids (aa), annotation, ortholog assignment and alignments. Appendix A gives statistics of reads, assembly metrics and the number of loci for phylogenetic analysis for each sample. Alignment summary statistics were calculated using AMAS [33]. Of an original number of 13,215 alignments, we filtered alignments with <100 aa aligned length and <10% variable sites, leaving 12,674 alignments with a total of 4,608,519 aa sites.

### 2.2. Phylogenetic Analyses of the Transcriptome Dataset

To reconstruct the backbone phylogeny for Terebelliformia, we used maximum likelihood and coalescent-based estimation methods (commands in Appendix A, all files have been deposited in a repository, see below). We first investigated the effects sequence representation across taxa in a concatenation framework by analyzing matrices with loci that were present in at least 17 of 21 taxa (80% occupancy, 1513 genes, 483,075 aa sites), and in at least 19 taxa (90% occupancy, 632 genes, 200,176 aa sites). We also investigated highly complete matrices, but with few genes, with 20 taxa (95% occupancy, 294 genes, 97,112 aa sites) and all 21 taxa (100% occupancy, 88 genes, 30,400 aa sites). The matrices were run as one partition with IQ-TREE 1.6.1 [34] on the CIPRES Science Gateway [35] under the LG+G4 model (the most frequently chosen model in 4614 of the 12,674 gene trees, see below), assessing support with 1000 ultrafast bootstraps [36].

Second, we made use of the large number of loci contained in the transcriptomes. Coalescent-based species tree approaches generally perform best on large sets of gene trees and remain accurate even if individual gene trees do not have full taxonomic coverage [37,38]. We therefore used all 12,674 loci for estimation of gene trees (mean number of sequences per alignment = 9.4, standard deviation = 4.8). For each alignment, we estimated the best-fitting aa substitution model with ModelFinder [39] and used the model chosen with the Bayesian Information Criterion to estimate the best tree with 1000 ultrafast bootstraps. The resulting gene trees were summarized into a species tree with ASTRAL-III v. 5.14.3 [40]. ASTRAL supports are local posterior probabilities, which are based on gene tree quartet frequencies and values above 0.95 are considered strongly supported [41]. 

A previous study on Terebelliformia mitochondrial genomes and three nuclear genes has identified compositional heterogeneity among taxa and suggested misleading effects on phylogenetic reconstruction [11]. We identified loci or gene trees with heterogeneity using three metrics. First, we excluded loci that violated the assumptions of substitution models that sequence evolution is stationary, reversible and homogenous, by using the matched-pairs test of symmetry [42] implemented in IQ-TREE v. 2 [43]. Second, we identified gene trees with heterogeneity in branch lengths that may cause long branch attraction, by measuring the standard deviation of long branch scores (LB_SD) [44] for each gene tree. Third, we excluded gene trees with low bootstrap support, which can indicate high gene tree estimation error, by averaging bootstrap values over each tree. The last two metrics were calculated in R (https://github.com/marekborowiec/metazoan_phylogenomics/blob/master/gene_stats.R). For each metric, we removed the extreme values (two standard deviations from the mean), which excluded between 77 and 562 gene trees (Appendix A) and summarized the remaining gene trees with ASTRAL. We also excluded all 1040 loci that were violating any of the three conditions and estimated a species tree from the remaining 11,634 homogenous gene trees. 

Another experiment targeted sequences that were potentially introduced from extraneous sources such as parasites, food or symbionts of the studied species or from cross-contamination during lab work or sequencing. We therefore excluded potentially problematic branches from gene trees and investigated whether the exclusion impacted the phylogenetic hypothesis presented here. The first approach removed identical sequences, which may have originated through spillover between samples. We identified 456 loci that had two or more identical sequences (824 total) and we removed the corresponding branches from the gene tree using pxrmt [45]. The second cleaning step aimed at unusually long branches, which would be expected if some sequences originated from a non-terebelliform parasite or commensal that is evolutionarily more distant than the ingroup. We used TreeShrink v. 1.3.3 [46] to remove terminals that had uncharacteristically long branches considering the species-specific distribution of branch lengths across all gene trees (6272 branches total). Altogether, 5.9% of all branches were removed, indicating that if contamination existed in the samples, it was at a low level. The “cleaned” gene trees were summarized with ASTRAL. 

### 2.3. Sanger Matrix and Morphological Characters

We included 121 terebelliform species (133 terminals) to study the relationships within the major clades and to understand character change across Terebelliformia. We used information from five Sanger-sequenced genes (2 mitochondrial regions; 3 nuclear regions) and 90 morphological characters. This dataset alone did not contain sufficient characters to resolve the deeper relationships. We therefore used the well-supported transcriptome phylogeny (80% matrix) with its 21 overlapping taxa as a backbone to constrain the relationships among the main lineages, while the remaining 112 terminals were free to be placed by the five gene and morphological partitions. This total evidence dataset therefore combines a backbone from high gene coverage with denser taxonomic coverage. 

A total of 315 new Sanger sequences were released to NCBI GenBank. Collection details, voucher information and accession numbers are given in Appendix A. Available sequence data from NCBI were included if the species had at least one mitochondrial and nuclear gene sequence available. Several potential new species are identified here and detailed taxonomic work is needed to formally describe these species. We use “cf.” to indicate tentative species identifications where the specimen in question matches the species description but was collected far from the type locality. For example, our specimen of *Thelepus* cf. *pascua* (Fauchald, 1977) matches the description for this species, but our specimen was collected in Indonesia while the type locality is in the Atlantic off Panama. We denote terminals as “sp.” where identification to the species level was not possible or specimens likely represent undescribed species because of unique traits. For instance, *Terebellides* sp. LA1 is an undescribed species from the southern Californian coast (SCAMIT, https://preview.tinyurl.com/tqayhud) and differs from most *Terebellides* by having separate branchial pairs with the branchial lamellae distant from each other (Figure 2J and Figure A3C). Similarly, *Octobranchus* sp. from California is similar to specimens referred to “*Bizzarobranchus*” or “*genus* B” (SCAMIT, https://preview.tinyurl.com/wnmmnv5), but has not been formally described.

DNA was extracted from specimens preserved in 95% ethanol using Qiagen DNeasy Blood and Tissue Kits (Qiagen Sciences, Germantown, MD, USA) following the manufacturer’s protocol. Gene regions were amplified using a PCR mixture of 12.5 µL GoTaq Green Master Mix (Promega, Fitchburg, WI, USA), 0.5 µL 25 mM MgSO_4_, 1 µL of each primer and 50–100 ng of DNA and ddH_2_O to yield a final reaction volume of 25 µL, or, when amplification failed, using PuReTaq Ready-To-Go PCR beads (GE Healthcare, Buckinghamshire, UK).

Partial fragments of the mitochondrial cytochrome c oxidase subunit I gene (COI; ca. 663 bp) were amplified with universal primers [47]. Partial fragments of the mitochondrial 16S rRNA gene (16S; ca. 540 bp) were amplified with different primer pairs [48,49,50], and ~1740 bp of the 18S rRNA gene (18S) was amplified with different primers [51,52,53]. Amplification of these three markers followed [21]. A fragment of the D1 region of the 28S rRNA gene (28S; ca. 350 bp) was amplified using the primers from [54] and following the protocol in [16]. A 327 bp fragment of histone H3 (H3) was amplified using the primers and protocol outlined in [55] and amplified as [16]. PCR products were purified with ExoSAP-IT (USB Corporation, Cleveland, OH, USA) and sequenced with the corresponding PCR primers. Overlapping fragments were merged into consensus sequences using Sequencher v. 4.10.1 (Gene Codes Corporation) or Geneious v. 7 (http://geneious.com). 

To integrate the 20 terebelliform species with sequenced transcriptomes into this matrix, the five genes were extracted from the transcriptome contigs using blastn v. 2.2.29+ (https://ftp.ncbi.nlm.nih.gov/blast/executables/blast+/2.2.29/) against a database of Sanger-sequenced sequences. We extracted the contigs with the highest similarity to specimens of the same species or to other related species. Contigs were trimmed after initial alignment to match the length of the Sanger sequences. 

Sequences were aligned on the MAFFT v. 7 online server [56,57] using default settings for the protein-coding coding COI and H3 genes, the iterative E-INS-i algorithm for 28S rRNA and 16S rRNA and the Q-INS-i algorithm that takes secondary structure into account for 18S rRNA. Final alignment lengths were COI: 516 bp, 16S: 667 bp, 18S: 2274 bp, 28S: 408 bp, H3: 327 bp.

Homologies of structures of the terebelliform body have been debated for decades. Different terms for potentially homologous structures are often used in each of the major groups. We provide a revision of the morphology of the terebelliform head, body and chaetae and our interpretations of their homologization (Appendix B). This resulted in a total of 90 morphological characters that were coded for all 133 terminals (state description in Appendix A). Morphological information was obtained from observation with a stereomicroscope. If characters could not be observed due to damage, as well as for species included from NCBI, the original descriptions were used and, if necessary supplemented from secondary studies [25,26,58,59,60]. Pale specimens were stained with Shirlastain A (SDL Atlas) or methyl green (Sigma-Aldrich, St. Louis, MO, USA) to enhance contrast. Chaetae were mounted on microscopic slides for examination with differential interference contrast optics.

### 2.4. Phylogenetic Analysis of the Total Evidence Matrix and Ancestral State Reconstruction

Molecular alignments and the morphological characters were concatenated, partitioned for each gene, and by codon position for the protein-coding genes (COI, H3). Matrices were analyzed with IQ-TREE v. 1.6 with the best fitting nucleotide substitution models and 1000 ultrafast bootstraps incorporating the constraint tree. Bootstrap support at constrained nodes was still free to be inferred from the total evidence matrix. Maximum likelihood (Mk1 model) ancestral state reconstruction of the morphological traits was investigated in Mesquite v. 3.31 [61]. All data files have been deposited in a repository (see below), commands are given in Appendix A.

## 3. Results

### 3.1. Phylogenomic Backbone Phylogeny

The transcriptomic dataset for the main lineages and *Arenicola marina* as an outgroup comprised 12,674 protein-coding genes and 4,608,519 amino acids. Concatenated matrices with taxon occupancies of 80% (17 taxa, 1513 genes, 483,075 aa) and 90% (632 genes, 200,176 aa) supported the same topology, as did the coalescent-based species tree estimation using all 12,674 gene trees, indicating a robust phylogenetic signal supporting the phylogeny (Figure 1). This coalescent-based species tree was stable when gene trees were subsampled to reduce heterogeneity in base composition, low bootstrap support and long branch gene trees (Appendix A). Exclusion of potentially extraneous sequences also resulted in the same topology with improved support on one node (Appendix A). Many nodes were supported with maximum support across all analyses. 

The phylogeny was often congruent with a traditional higher-level classification of Terebelliformia of Grube, 1850, Malmgren, 1866 and Hessle, 1917, with a few exceptions. The monophyly of Trichobranchidae, Alvinellidae and Terebellidae was confirmed (Figure 1). We found Ampharetidae, traditionally categorized as Ampharetinae and Melinninae, to be polyphyletic. Melinninae was sister to Terebellidae, with high, albeit not maximum support. Support for this relationship was high when all gene trees were analyzed (PP = 98) and using the 80% matrix (bs = 97). With fewer genes using the 90% matrix, bootstrap support dropped (bs = 56) and even smaller matrices (>95% occupancy) resulted in *Melinna* shifting its position to being the sister to Trichobranchidae and Terebellidae (Appendix A). On those small matrices, support for a clade of *Melinna*, Trichobranchidae and Terebellidae was moderate to high (bs = 64–92), but support for Terebellidae plus Trichobranchidae was low to moderate (bs = 32–56). In all other trees that were built from greater amounts of data, as well as the subsetted coalescent-based species trees, Melinninae was the sister group to Terebellidae with high support (PP = 98–99, Appendix A). Because *Melinna* did not group with Ampharetinae, Melinninae now becomes Melinnidae.

Because Ampharetidae sensu lato was polyphyletic, it is restricted here to the former Ampharetinae, which henceforward becomes Ampharetidae. This clade was the sister group to Alvinellidae. Within Ampharetidae, three clades were identified and their naming was justified by the denser sampling as discussed below, Ampharetinae (sensu stricto), Amaginae (new status as subfamily, see below) and Amphicteinae (new status as subfamily, see below). Relationships were maximally supported in the coalescent-based species tree but the node supporting Ampharetinae as the sister to Amaginae and Amphicteinae was lower in the 80% concatenated matrix (bs = 95) and in the 90% concatenated matrix (bs = 66). Interestingly, the smaller 95% and 100% matrix had better support for this relationship (bs = 87–90, Appendix A). Within Terebellidae, Thelepodinae was found to be the sister group to all the remaining Terebellidae across all analyses, which is referred to as Terebellinae. The inclusion of Polycirridae in Terebellinae was consistently found with full support, warranting the subdivision of Terebellinae into several clades, namely Polycirrini, Terebellini, Procleini and Lanicini (ranked as tribes, see below). In Trichobranchidae, *Trichobranchus roseus* was placed as the sister group to *Terebellides* sp. in all analyses. Alvinellidae are currently considered to contain two genera, but *Paralvinella* was found to be paraphyletic with *P. irlandei* (now *Nautalvinella*) as the sister to a clade of *Alvinella* and the other *Paralvinella* species with high, but not maximum support (bs > 95, PP = 91). 

### 3.2. Total Evidence Dataset

The phylogenetic tree from ML analysis of the matrix consisting of five genetic markers and 90 morphological characters (4249 characters total) informed mostly on shallower taxonomic levels, while the deeper nodes were poorly resolved and some were in conflict with the transcriptome phylogeny (Appendix A). Because some lineages were constrained to match the transcriptome topology (Figure 1), we only report relationships within the main lineages here. 

Ampharetidae (excluding Melinnidae) consisted of three major clades (Figure 2) that we apply names to as follows: a diverse clade of Ampharetinae (sensu stricto, with the type genus *Ampharete* Malmgren, 1866); Amphicteinae Holthe, 1986 (formerly a tribe Amphicteini, with the type genus *Amphicteis* Grube, 1850 and taxa included = *Amphicteis, Hypania* and *Noanelia*); Amaginae Holthe, 1986 (formerly a tribe Amagini, with the type genus *Amage* Malmgren, 1866 and taxa included = *Amagopsis, Amage* and *Amphisamytha*). In Melinnidae, *Isolda* species were highly supported as the sister to *Melinna*. Their sister group was *Samythella neglecta*, which is normally considered to be placed within Ampharetinae. In the total evidence matrix, this relationship with Melinnidae received moderate support (bs = 77), while it was highly supported when molecular data was analyzed alone (bs = 97, Appendix A). The sequences for this species come from a previous study, where it was placed as the sister group to Alvinellidae and the three ampharetid clades described here [2]. We could not investigate the specimen and coded the morphological characters from the literature, which placed it in a more expected position among some Ampharetidae (Appendix A). 

Terebellidae contained Thelepodinae and Terebellinae (Figure 2). Within Thelepodinae, the representative of Telothelepodidae, *Rhinothelepus lobatus,* was nested within Thelepodinae as the sister to *Thelepus* cf. *pascua* (bs = 98). As in the transcriptome topology, the inclusion of Polycirridae within Terebellinae was also supported by the unconstrained total evidence matrix (bs = 86, Appendix A) but opposed by the morphological partition alone (Appendix A). Terebellinae therefore contained a number of clades, which we refer to as tribes introduced by Holthe [25]. Polycirrini Holthe, 1986, included *Amaeana*, *Hauchiella*, *Lysilla*, *Polycirrus* and *Biremis* and relationships among them were generally well-supported. Polycirrini was sister to Terebellini, which contained the type taxon of Terebellidae, *Terebella lapidaria* Linnaeus, 1767. Terebellini further included members of *Artacama, Leaena, Neoamphitrite, Lanassa* and *Spinosphaera*. The sister to Terebellini+Polycirrini was a clade of *Eupolymnia* (including *Reteterebella*), *Laphania* and *Proclea,* which we refer to as Procleini Holthe, 1986, with type genus *Proclea* Saint-Joseph, 1894. The sister to Terebellini+Polycirrini+Procleini was a group for which we use the name Lanicini Holthe, 1986, with type genus *Lanice* Malmgren, 1866 and further includes *Loimia, Axionice, Lanicola, Pista* and *Scionella*.

In Trichobranchidae, *Octobranchus* sp. (“*Bizzarobranchus*”) was the sister group to *Octobranchus lingulatus*. Together they were the sister to various *Terebellides* species. The undescribed *Terebellides* sp. (LA1), which has distantly spaced branchial lamellae (Figure 2J) was the sister group to *Terebellides* with fused branchiae. *Trichobranchus* had a long branch and contained two main clades. In Pectinariidae, *Petta* was sister to *Pectinaria*. In Alvinellidae, many species were constrained by the transcriptome topology but the paraphyly of *Paralvinella*, with *Paralvinella irlandei* and *P. pandorae* being the sister to *Alvinella* and other *Paralvinella* species, was supported even by the unconstrained molecular and morphological matrices (Appendix A). *Paralvinella irlandei* and *P. pandorae* are often considered subspecies of *P. pandorae* but we treat them as separate species based on their branch length differences (Figure 2). We refer these two taxa to *Nautalvinella* Desbruyères and Laubier, 1993, a name formerly with the rank of subgenus within *Paralvinella.*

We found several paraphyletic genera (in addition to the paraphyly of *Paralvinella*). In Lanicini, the type species *Lanicola lobata* Hartmann-Schröder, 1986 occurred in a different clade than *L. carus* (Young and Kritzler, 1987), which was originally described as the type of *Paraeupolymnia*. We therefore refer to the latter as *Paraeupolymnia carus* Young and Kritzler, 1987. In the other paraphyletic genera, we refrain from taxonomic changes until further evidence is available. In Terebellini, the type species *Neoamphitrite affinis* (Malmgren, 1866) was in a different clade than *N. robusta* (Johnson, 1901). In Procleini, *Eupolymnia* was paraphyletic with respect to *Reteterebella*, but we were missing type species (*Eupolymnia nesidensis* (Delle Chiaje, 1828) and *Reteterebella queenslandia* Hartman, 1963), which would be necessary to change the nomenclature. In Polycirrini, *Polycirrus arcticus* grouped outside of other *Polycirrus* species but we did not include the type species *P. medusa* Grube, 1850. In Thelepodinae, both *Thelepus* (including the type species *Thelepus cincinnatus* (Fabricius, 1780)) and *Streblosoma* species appeared in multiple parts of the phylogeny. For taxonomic issues involving previously published sequences, we could not investigate all specimens to assess if they should be placed in new or existing genera. This was the case in Ampharetinae, where the type species *Neosabellides elongatus* (Ehlers, 1913) did not form a clade with *N. lizae* Alvestad and Budaeva, 2015, but we could not evaluate a specimen of the latter. Further, the type species *Pista cristata* (Müller, 1776) was in a different clade than *P. australis* Hutchings and Glasby, 1988, but both came from previous studies [2,62]. 

### 3.3. Character Evolution

Ancestral state reconstruction supported a terebelliform ancestor that possessed segmentally arranged branchiae, notochaetae, neurochaetae and was tube-dwelling. Reconstruction of the number of branchial pairs suggests that the plesiomorphic condition was four pairs of branchiae (likelihood l = 0.998, Figure 3). This state was retained in Pectinariidae, Melinnidae and Alvinellidae. It was the ancestral state of Ampharetidae and Trichobranchidae, but branchial pairs were lost in multiple lineages. Multiple transitions from segmentally arranged branchiae to anteriorly telescoped branchiae that appear to lie on the same segment happened in Ampharetidae+Alvinellidae, Melinnidae, and, in an extreme form in *Terebellides* with a fused branchial stalk (excluding *Terebellides* sp. LA). The most recent common ancestor of Terebellidae was reconstructed to have had three pairs of branchiae (l = 0.919), with complex patterns of reductions and one gain of additional branchial pairs in *Streblosoma pacifica*. Complete loss of branchiae occurred at least six times in Terebellidae (Figure 3). The firm placement of Polycirrini within Terebellinae implies that the loss of branchiae, often accompanied by loss of notochaetae and neurochaetae, were secondary reductions. The loss of notochaetae happened at least twice in Polycirrini, in *Biremis* cf. *blandi* and in *Hauchiella renilla*, while the loss of neuropodia was only observed once in the ancestor of *Hauchiella* and *Lysilla* cf. *pacifica* (Figure 3). 

The placement of Melinnidae as the sister group to Terebellidae resulted in several former proposed apomorphies for Alvinellidae+Ampharetidae (in the traditional sense for the latter, i.e., including Melinninae) being reconstructed either as plesiomorphic for Terebelliformia or as independent gains (Figure 4). This includes the hood-like prostomium of Ampharetidae, Alvinellidae and Melinnidae that was reconstructed as ancestral for all Terebelliformia except for Pectinariidae (l = 0.982). The pectinariid head is thought to be a fusion of prostomium and peristomium [28,63] and the trait could therefore not be evaluated in the group. In the present scenario, the hood-like prostomium transformed into the narrow ridge of many Terebellidae and to the expanded sheet-like prostomium of Trichobranchidae. The attachment of the palps outside of the mouth was reconstructed as the plesiomorphic state in Terebelliformia (l = 0.953), with independent transitions into the mouth in Ampharetidae+Alvinellidae and in Melinnidae. Again, the interpretation of the head region in Pectinariidae is important. Their tentacles are outside of the mouth, but because of the fusion of peristomial and prostomial parts their structural origin is currently unknown.

In Ampharetidae, most characters were largely ambiguous for supporting clades, including the commonly used taxonomic characters (number of branchiae, presence of paleae, number of chaetigers, presence of modified notopodia [64]). An exception was a clade defined by pinnate palps (*Ampharete, Asabellides, Neosabellides, Sabellides*) except for *Neosabellides lizae*, which has reduced the pinnate palps under the present scenario (Figure 4). 

In Terebellidae, Thelepodinae have the apomorphies of filamentous branchiae and a knob-like subrostral process on their neurochaetae that points forward. Terebellinae are characterized by uncini in double rows, which was reversed to single rows in those Polycirrini that have neurochaetae. Polycirrini are united by the lack of branchiae and the expanded prostomium (convergent with Trichobranchidae). We could not identify an autapomorphy for Terebellini, nor for Procleini based on the characters used here. Members of Lanicini share large lateral lappets on their anterior bodies (Figure 2H). 

## 4. Discussion

We summarize the nomenclatural changes supported by our study in Figure 5 and compare them to the two currently prevailing systematic systems [13,25]. The most significant change from our study is the finding that Melinnidae are not part of Ampharetidae, but instead are more closely affiliated with Terebellidae. Two previous studies focusing on ampharetid relationships found Melinnidae as the sister group to Alvinellidae and Ampharetinae, but their rooting on Terebellidae did not allow for the *Melinna*+Terebellidae relationship to be recovered [2,21]. Studies that included more lineages in Terebelliformia did not place melinnids confidently [16] or found an inclusion in Ampharetinae based on combined morphological data and the 28S gene [20], while morphology alone kept Melinnidae in the traditional position [14]. The transcriptome data strongly supported *Melinna oculata* as the sister group to Terebellidae, separate from Ampharetidae. These groupings suggest that many of the characters previously considered apomorphic for Ampharetidae may be symplesiomorphic for Terebelliformia, or have independent origins. This includes the four pairs of digitate branchiae that are now supported as the ancestral state of Terebelliformia and the hood-like prostomium which is plesiomorphic for Terebelliformia except Pectinariidae. Interestingly, Holthe [25] already suggested an “archaeoterebellomorph” with four pairs of branchiae and a large prostomium. 

The placement of *Samythella neglecta* (Ampharetidae) as the sister to Melinnidae warrants further investigation. The sequences came from a previous study that supported the position *S. neglecta* as the sister group to other Ampharetinae and Alvinellidae [2], which is not the expected position from morphology but less surprising than the present relationship. It is possible that the broader sampling used here and the different rooting resulted in the changed position of *Samythella neglecta*. We suggest not to include this taxon in Melinnidae until further evidence is available. 

Aside from the placement of Melinnidae, the transcriptome and total evidence data were in agreement with the historical delineation of the major clades. In particular, we found evidence for monophyletic Pectinariidae, Trichobranchidae, Ampharetidae (excluding Melinnidae), Alvinellidae and Terebellidae. The monophyly of Terebellidae contradicts the morphology-based proposal that Terebellidae includes all other Terebelliformia [13]. There was strong support for Thelepodinae and Terebellinae, which in turn contained the clades Lanicini, Procleini, Terebellini and Polycirrini. We found no evidence for the proposed separation of Polycirridae from Terebellidae [13] but rather support their inclusion within Terebellinae, as the sister group to Terebellini. Polycirrini are therefore simplified terebellids that have lost their branchiae, reversed the double rows of uncini that characterize Terebellinae and reduced chaetae in some species (Figure 3 and Figure 4). Their affiliation with Terebellidae was previously suggested because they all share palps inserting outside of the mouth and their glandular shields [24,28]. Phylogenetic analyses based on morphological data found Polycirrini inside of Thelepodinae [23], as the sister group to Terebellinae+Thelepodinae [24], or as the sister group to all other Terebelliformia [13]. The latter result we also recover when the morphological partition is analyzed on its own (Appendix A). It is possible that the high degree of non-applicable morphological characters due to the absence of branchiae and often also of chaetae produced these placements of Polycirrini, which differ markedly from the hypothesis strongly supported by both Sanger-sequenced genes and transcriptomic data. The other tribes of Terebellinae, Procleini or Terebellini, were not identified by clear apomorphies based on the present data. Lanicini were characterized by large lappets behind their heads. Such a grouping has been made ad hoc before [65] and is confirmed here with strong support based on molecular data. 

Our study included only one member of Telothelepodidae, a group that is currently viewed as separate from Thelepodinae because of their elongate, narrow upper lip [13,66]. Although transcriptomic data is not yet available for any member of Telothelepodidae, the total evidence analysis confidently placed *Rhinothelepus lobatus* inside Thelepodinae. The species was deeply nested within several members of *Thelepus* and *Streblosoma* and was found the closest relative to *Thelepus* cf. *pascua*, which lacks the elongate, narrow upper lip. This suggests that at least *Rhinothelepus* is part of Thelepodinae. The other 11 species of Telothelepodidae [66] need to be included in future molecular studies. 

Within Ampharetidae, our sampled species fell into three clades, Amphicteinae, Amaginae and Ampharetinae. We built on data from a previous study on Ampharetidae, which found the same three corresponding clades, but resolved different relationships among them [2]. Ampharetinae is diverse and additional sampling will be required to resolve the relationships further. At this stage, we could not identify traits that supported specific clades, except for Ampharetini Holthe, 1986, of which we sampled *Ampharete, Asabellides, Neosabellides* and *Sabellides*, which share pinnate buccal palps. The exception is *Neosabellides lizae*, which has smooth buccal tentacles [67], a loss given the present phylogeny. There are several more genera that have pinnate palps [68], which also should be included in future molecular studies. Holthe [25] recognized other tribes of Ampharetini, for which we had no (Melinnampharetini) or too few species sampled to assess their validity (Samythini, Lysippini and Auchenoplacini). Our data indicate that Sosanini Holthe, 1986 is not valid because at least two of the included genera, *Sosane* and *Anobothrus*, are in different parts of the tree. This clade was defined based on the presence of paleae and the elevated notopodia of the posterior thorax [25]. Both characters were reconstructed to have originated at least twice using our data.

We found that Alvinellidae were sister to Ampharetidae and, for nomenclatural stability, we retain the family rank. Upon their discovery, alvinellids were placed inside Ampharetidae as a subfamily but later separated [17,18] and this status is now well accepted. The transcriptomic and Sanger sequences found that *Paralvinella* is paraphyletic owing to the position of *Alvinella*. This has been suggested based on a previous study using transcriptomic data, where *Alvinella* and *Paralvinella* were not reciprocally monophyletic in all gene trees [69], but an outgroup was missing to fully interpret these results. Here, we showed unequivocally that *Paralvinella pandorae* and *P. irlandei* were a sister group to *Alvinella* and the other *Paralvinella* species and so should be referred to *Nautalvinella*, originally a subgenus of *Paralvinella* that was erected based on gill morphology [70].

We further suggest resurrecting *Paraeupolymnia* because *Lanicola lobata* Hartmann-Schröder, 1986, the type taxon of *Lanicola*, was distinctly separate from *L. carus* (Young and Kritzler, 1987), which was originally described as the type taxon of *Paraeupolymnia*. We resurrect *Paraeupolymnia carus* based on our findings (reversing the opinion in [71]). 

Further investigation is needed on several other paraphyletic genera for which we currently have insufficient information to make nomenclatural changes. *Pista cristata* (Müller, 1776) is a type of *Pista*, a genus and species that is suspected to contain many independent lineages [65]. We show here that at least *P. australis* may need to be reassigned. The sequences from this individual came from another study [62] and we were not able to investigate if the species should better be referred to a different genus or possibly to *Paraeupolymnia*, as it was the sister to *P. carus.* The type species *Neoamphitrite affinis* (Malmgren, 1866) was not monophyletic with *Neoamphitrite robusta* (Johnson, 1901), which indicates that the latter may have to be moved to a different genus. *Reteterebella* nested in *Eupolymnia* but the respective type species, *Eupolymnia nesidensis* (Delle Chiaje, 1828) and *Reteterebella queenslandia* Hartman, 1963, were not included here. The type species *Neosabellides elongatus* (Ehlers, 1912) and *Neosabellides lizae* Alvestad and Budaeva, 2015, the sequences for which came from a different study [2], were not monophyletic in Ampharetini, which indicates that the latter may have to be referred to a different genus. We found *Thelepus* and *Streblosoma* to be paraphyletic. The type species *Thelepus cincinnatus* has been recently redescribed and split into four species in the northern Atlantic based on morphology [72] and its position in our analysis indicates that both *T.* cf. *pascua* and *T. verrilli* should be referred to different genera. As we did not include the type species *Streblosoma bairdi* (Malmgren, 1866), the status of *Streblosoma* cannot be resolved but the position of the type species *T. cincinnatus* indicates that *S. kaia* Reuscher, Fiege and Wehe, 2012 may have to be included in *Thelepus*. *Polycirrus arcticus* grouped with *Biremis* cf. *blandi*, not with the other *Polycirrus* species sampled here, but we did not include the type species *P. medusa* Grube, 1850 and thus cannot resolve the delineation of *Polycirrus*. These findings may not be surprising given that morphological analysis indicated that taxonomic issues existed in *Polycirrus, Lysilla* and *Amaeana* [73]. These findings highlight the crucial need for detailed taxonomic revisions incorporating molecular data and broad sampling. 

## 5. Conclusions

Our study demonstrated how the integration of multiple approaches to reconstruct phylogenetic hypotheses from transcriptomics, to Sanger sequencing of a few phylogenetically informative genes, over morphological analysis can resolve problematic phylogenies, while optimally using available material and resources. Our results have implications on the taxonomy and systematics of the group and future revisions of potential new species are necessary. Our findings also impact our understanding of the evolution of the complex branchiae and chaetae within Terebelliformia and show that loss and transformation are widespread.

## Figures and Tables

**Figure 1 biology-09-00073-f001:**
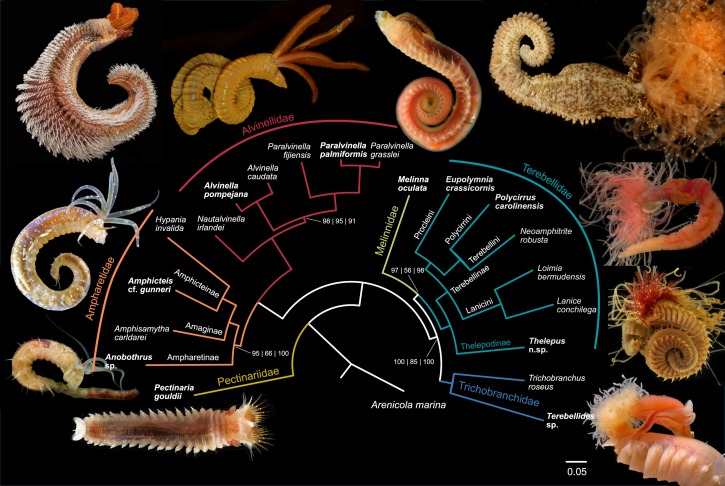
Relationships of the main groups of Terebelliformia based on transcriptomic data. The topology and branch lengths are from maximum likelihood inference on the supermatrix with 80% taxon occupancy (≥17 taxa, 1513 genes, 483,075 aa). The same topology was also found with 90% taxon occupancy, with all 12,674 gene trees summarized with ASTRAL, and when using subsets of the gene trees to reduce heterogeneity. Only nodes having less than full support in at least one of the analyses were annotated with support values, all others had full confidence across analyses. Support values are: bootstrap 80% matrix | bootstrap 90% matrix | ASTRAL local posterior probabilities. Photos correspond to the closest terminal in bold.

**Figure 2 biology-09-00073-f002:**
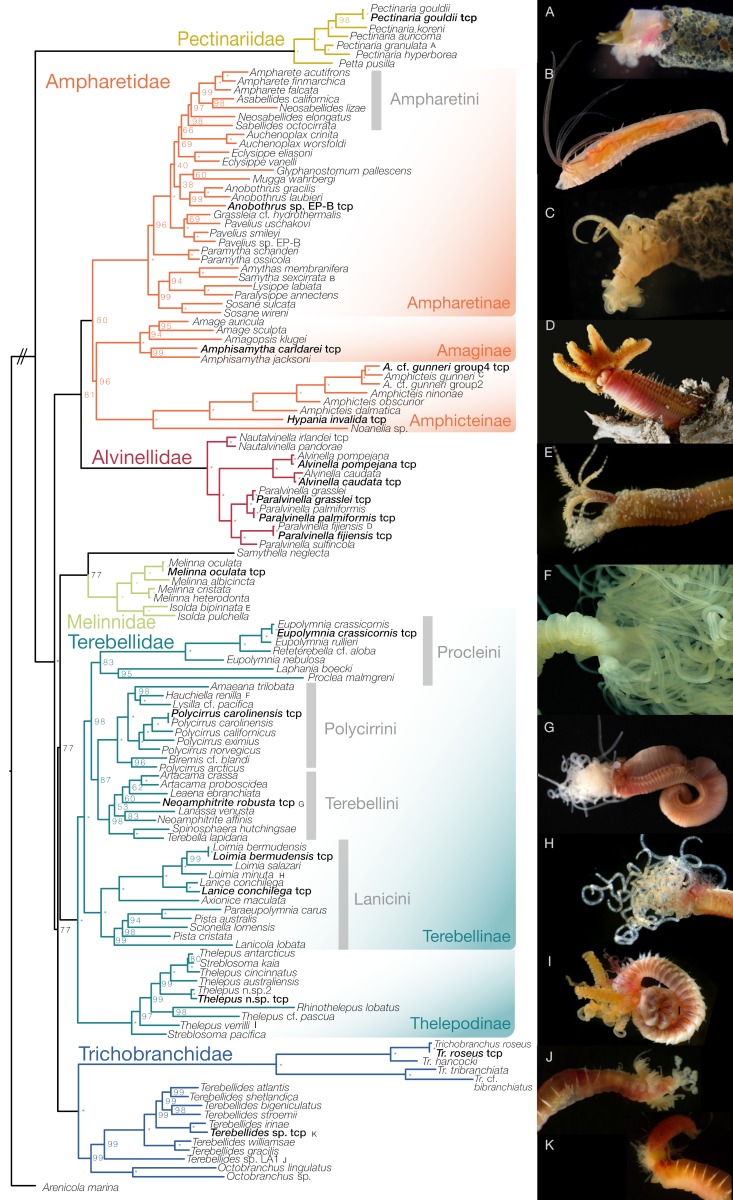
Relationships among Terebelliformia based on the multigene and morphological dataset (4249 characters) using maximum likelihood and constrained to the transcriptome backbone. Note that bootstrap support was free to vary despite the topological constraint. Asterisks indicate maximum bootstrap support. Terminals in bold are shared with the transcriptome phylogeny. The animals shown on the right are labeled in the phylogeny with letters (**A**–**K**). (**A**) *Pectinaria granulata* SIO-BIC A9441; (**B**) *Samytha sexcirrata* SIO-BIC A1110; (**C**) *Amphicteis* cf. *gunneri* group2 SIO-BIC A1107; (**D**) *Paralvinella fijiensis* SIO-BIC A8627 (not the sequenced specimen); (**E**) *Isolda bipinnata* SIO-BIC A9437; (**F**) *Hauchiella renilla* SIO-BIC A1116; (**G**) *Neoamphitrite robusta* SIO-BIC A9453; (**H**) *Loimia minuta* SIO-BIC A9452; (**I**) *Thelepus verrilli* SIO-BIC A9462; (**J**) *Terebellides* sp. LA1 SIO-BIC A9464; (**K**) *Terebellides* sp. SIO-BIC A9465.

**Figure 3 biology-09-00073-f003:**
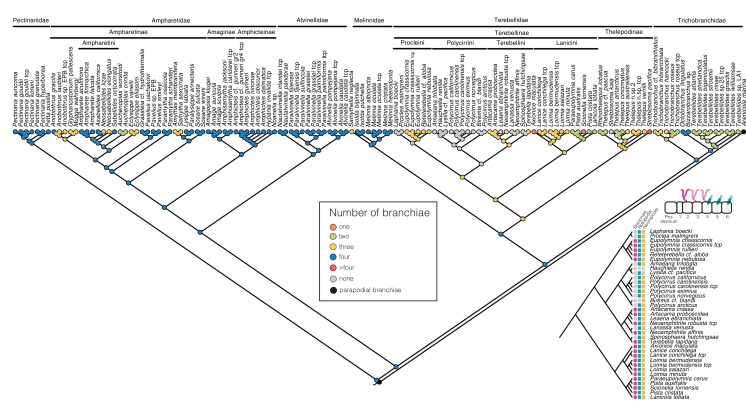
Evolution of the number of branchiae and chaetae in Terebelliformia. Maximum likelihood character reconstruction of the number of branchial pairs. Inset: Mapping of the presence (colored squares) or absence (gray squares) of branchiae, notochaetae and neurochaetae in Terebellinae showing complex patterns of losses.

**Figure 4 biology-09-00073-f004:**
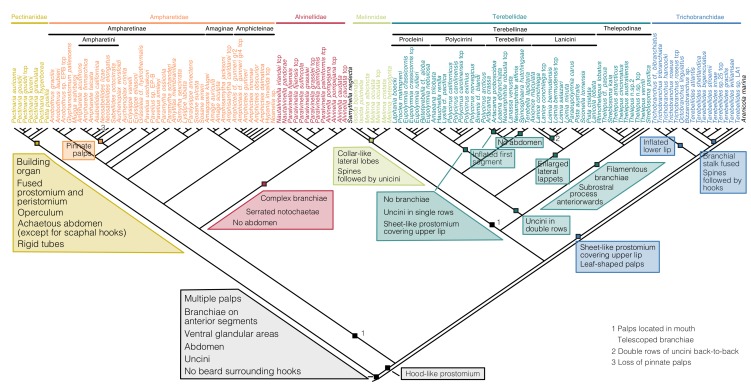
Selected apomorphies of Terebelliformia.

**Figure 5 biology-09-00073-f005:**
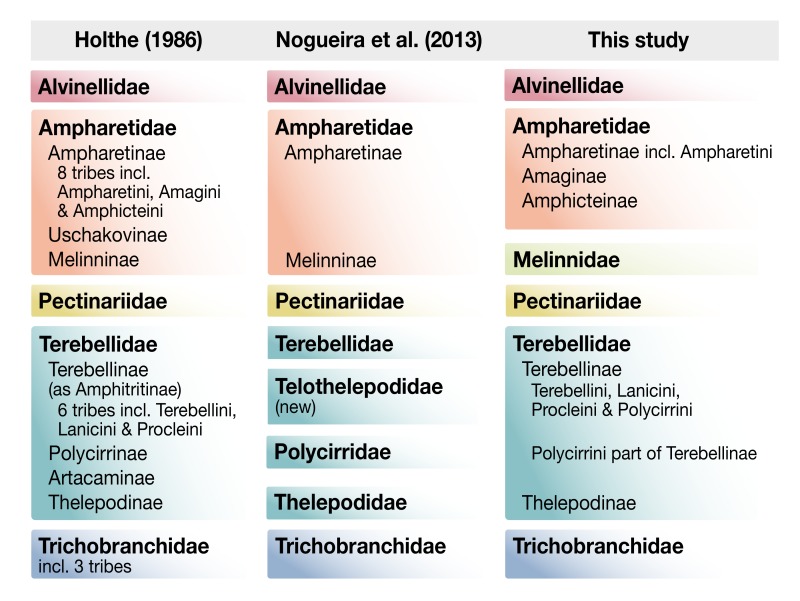
Summary of nomenclatural changes of this study in relation to two previous works.

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
