# Peer review of "Spaghetti to a Tree: A Robust Phylogeny for Terebelliformia (Annelida) Based on Transcriptomes, Molecular and Morphological Data"

_biology, 2020, doi:10.3390/biology9040073_

Round 1

Reviewer 1 Report

This is a detailed description of the phylogenetics of a little-known group, using 20 transcriptomes (12,674 orthologs) to build a backbone phylogeny, then 2 mtDNA genes and 3 nuclear genes across 121 species to flesh this tree out. The combination of molecular phylogenetics and thorough knowledge of morphology are a strength.

Personally, I would not use the 90 morphological characters in the phylogenetic analysis though. The paper essentially tests the taxonomy of the group using independent molecular characters. The current taxonomy is a direct reflection of morphological characters already, so there is a degree of circularity here, especially when you then map morphological characters onto the tree (or are characters in Figs. 3-4 excluded from the phylogenetic analysis?). Having said that, the molecular characters (98%?) probably swamp the morphological characters, but it very much depends on how many are phylogenetically informative, and the distribution of their character states.

The writing does get a bit dense and descriptive, particularly as it relates to taxonomy. It would help if some of this could be reduced, or perhaps structured with subheadings to make it more accessible? Some of the writing is a bit casual or careless, but only in a minor way: taxa are always singular, and taxa need to be distinguished from their members. I make some suggestions below, but it could benefit from a read through by a systematist to formalize some of the language.

Specifics:

20  Suggest saying actual rank, rather than “group within” (e.g. “subfamily of”)

36  “OF annelid”?

37  Taxa are always singular, so say “members of” for example.

39  As above- reword to distinguish between talking about a taxon and its members.

41  “continually” might be better.

58  Delete: “on”

73  Colon break or new sentence needed at “other data”, or re-word.

102  “total evidence approach” may be a bit misleading here, as it might suggest using morphological data (and maybe others) to create the phylogeny, whereas I don’t think you mean that.

113  The transcriptome approach is a little light on replicate taxa. It is hard to make reliable generalisations and re-classification of Melinninae with just the one taxon, as we don’t know whether it represents an anomaly at the level of the entire subfamily, or just one particular species. The finer mtDNA analysis ameliorates this problem.

132  “positions were”?

153  “NUMBER of loci”

155  Might be a good idea to check that the phylogeny favoured is the same for a consistent dataset. Looks like this is robust from the Results section. Need to give some values for the amount of missing data.

172  “excluded”?

263  Looking at the tree, would it not be better to say that Ampharetidae, as currently described, is POLYphyletic? That is, you have not labelled ancestral taxa leading to Melinninae in orange.

322  Better to say: “placed in the Ampharetinae”.

Reviewer 2 Report

The paper of Stiller et al. is focused on the elucidation of phylogenetic relationships of the ‘spaghetti worms’ (Terebelliformia) based on the analyses of the transcriptomic data supplemented by a much larger dataset of selected terebellid genes and morphological characters. 

I consider the paper of a very good scientific quality with concise hypotheses and conclusions based on properly performed comprehensive phylogenetic analyses.

I only have minor comments/suggestions:

  1. Introduction, Line 91: "Terebelliforms may have been ancestral “naked” with poorly developed chaetae and were free-living". The life history has not been discussed anywhere in the introduction as well as you do not investigate this later. Even though I believe the evolutionary history of the organism's lifestyles is very interesting, I suggest omitting the second part of the sentence unless you don't feel it is important for your paper.
  2. M&M, Line 134: How were the potentially present contaminant sequences, i.e. of environmental origin (e.g. from ingested food) or by lateral gene transfer) identified and removed? If not removed from your dataset, can these data influence the results of your phylogenetic analyses? Include the section on the identification/removal of such sequences in this M&M. 
  3. M&M, Line 209: The sentence seems unfinished.
  4. Results, Line 269: Bs=64 is not high and later you refer to bs=66 as "it lost support" (lines 288-289). Be consistent.
  5. Results, multiple lines: I am confused by the use of Melinnidae/Melinninae throughout the text of the MS. You said that “Ampharetidae is traditionally categorized as Ampharetinae and Melinninae” and that Ampharetidae was paraphyletic (Ampharetinae and Melinninae did not cluster together). Because Ampharetidae sensu lato was paraphyletic, it is restricted here to the former Ampharetinae, which hence becomes Ampharetidae." Does the same situation apply for Melinninae/Melinnidae? At line 264, you use the term Melinninae but at line 271 and below as well as in Fig. 1. you use Melinnidae. This is also confusing in the abstract. Clarify, please.
  6. Results, Line 262: Include also the reference for the results of the phylogenetic analysis.
  7. Figure 1: The picture is beautiful but I suggest including the legend for the names of the pictured organisms in the same way as done in Fig.2.
  8. Figure 2: Maximum nodal support is probably shown by the asterisk in the tree but this should be explained in the figure legend.
  9. Figure 2: "tcp denotes terminals which are shared with the transcriptome phylogeny." It would be better visible if you label these taxa in bold in the tree.
